# Transmit-Array, Metasurface-Based Tunable Polarizer and High-Performance Biosensor in the Visible Regime

**DOI:** 10.3390/nano9040603

**Published:** 2019-04-11

**Authors:** Kai He, Yidong Liu, Yongqi Fu

**Affiliations:** School of Physics, University of Electronic Science and Technology of China, Chengdu 610054, China; someshaxiong@gmail.com

**Keywords:** metamaterials, metasurfaces, biosensors, polarizers, splitter, Mie resonances

## Abstract

There are two types of metasurfaces, reflect-array and transmit-array,—which are classified on the basis of structural features. In this paper, we design a transmit-array metasurface for *y*-polarized incidence which is characterized by having a transmission spectrum with a narrow dip (i.e., less than 3 nm). Furthermore, a tunable polarizer is achieved using linear geometric configurations, realizing a transmittivity ratio between *x*- and *y*-polarized incidence ranging from 0.031% to 1%. Based on the narrow-band polarization sensitivity of our polarizer, a biosensor was designed to detect an environmental refractive index ranging from 1.30 to 1.39, with a factor of sensitivity *S* = 192 nm/RIU and figure of merit (*FOM*) = 64/RIU. In the case of a narrow-band feature and dips in transmission spectrums close to zero, *FOM** can have a value as large as 92,333/RIU. This unique feature makes the novel transmit-array metasurface a potential market candidate in the field of biosensors. Moreover, transmit-array metasurfaces with lossless materials offer great convenience by means of detecting either the reflectance spectrum or the transmission spectrum.

## 1. Introduction

Metasurfaces consist of periodic unit cells of one subwavelength thickness and allow for arbitrary manipulation of the amplitude, phase, and polarization of incident light [1,2]. At present, metasurfaces have been applied in nanophotonic devices, such as holograms [3,4,5] and planar lenses [6,7,8,9], and also surface wave manipulation [10,11,12,13,14,15,16,17,18]. Plasmonic metasurface devices permit light modulation over the scale of one wavelength, where the intrinsic ohmic loss of metal poses a severe issue, especially in optical wavelengths, greatly limiting the performance of plasmonic devices. However, low-loss and high-refractive-index dielectric resonators have attracted much attention due to their ability to address metallic counterpart energy losses and support strong electric and magnetic mode resonances—defined as Mie resonances—in the optical regime [19,20,21,22,23,24,25]. Under the effect of Mie resonances of dielectric materials and given that the period of arrays with lossless materials are shorter than one working wavelength—to prevent the generation of grating diffraction—scattering light is allowed to propagate along the forward (normally transmitted) or backward (normally reflective) direction through tuning of the geometric parameters [25]. Polarizers, which are widely used optical devices, show different reflectance and transmittivity for *x*- and *y*-polarized incident light. Gallium nitride (GaN), a III-V semiconductor material, is lossless in the visible regime and has also been well applied as an active material in light-emitting diodes (LEDs) [26,27]. In this paper, we designed a tunable polarizer consisting of GaN scatterers distributed onto SiO_2_ substrate with a tunable transmittivity ratio between *x*- and *y*-polarized incident light for the wavelength of 532 nm. We define the above ratio as *R_t_* = *T_y_*/*T_x_*, where *T_y_* and *T_x_* represent the transmittivity with *x*- and *y*-polarized incident light, respectively. We have achieved a tunable polarizer with *R_t_* ranging from 0.00031 to 1. When *R_t_* is set to its lowest value, we can achieve perfect reflection [28] in narrow-band wavelengths smaller than 3 nm for visible wavelengths with transmit-array metasurfaces. Using the above narrow-band polarization sensitivity feature, we can design a biosensor with transmit-array-based metasurfaces to detect the refractive index variation of the surrounding environment. Most researchers studying the detection of environmental refractive index are exploring metasurfaces working in near-infrared wavelengths, and reflective structures for preventing transmission that are characterized by bottom metals with a thickness much larger than skin depth [29,30,31,32,33,34,35]. Recently, a novel biosensor was presented in the literature [18], where the environmental refractive index was detected by the transmission spectrum. Another biosensor was presented which uses both the reflectance and transmission spectra [36]. However, both of the reported structures contain metal parts, which can cause energy absorption. Our counterpart is structured with pure lossless dielectric materials and without metals. An apparent advantage of our design is that it does not produce heat dissipation caused by the inherent ohmic loss in metals. We can detect either the minimum value of the transmission spectrum or the maximum value of the reflectance spectrum, which offers great flexibility. As a result, our biosensor has good performance in the narrow-band transmission spectrum and a factor of sensitivity *S* = 192 nm/RIU when detecting a large range of refractive indexes, *n*, ranging from 1.3 to 1.39. Benefiting from the distinct narrow-band transmission spectrum in the environment and an extremely low transmittivity of 0.06% in resonant wavelengths with different refractive index, the figure of merit (*FOM*)*—one practical factor used to evaluate the performance of biosensors, which is concerned about the intensity variation due to the refractive index variation—can have a value as large as 92,333/RIU. With an environmental refractive index of *n* = 1.33, we determined the sensitivity of our biosensor using thin layers of various thicknesses and *n* = 1.5. For an incident wavelength of 504 nm, thin layers with various thicknesses ranging from 40 to 90 nm produced different transmittivity values ranging from 0.76 to 0.008. When the thickness of the thin layer was constant, at 90 nm, its refractive index ranged from 1.5 to 1.7, and had a high *FOM** value of 15,167/RIU. Consequently, we present a new method to design high-performance biosensors by utilizing transmit-metasurfaces with narrow-band polarization dependence.

## 2. Materials and Methods

When unit cells of metasurfaces are designed with different sizes in the direction of incident polarizations, metasurfaces possess different transmission coefficients for different incident polarizations. For our unit cells, we employed rectangular posts in the design of scatterers, as shown in Figure 1. The period *P* of a unit cell was the same in both *x* and *y* directions. Based on the geometrical symmetry of rectangles, the polarization state of the reflected or transmitted light does not deflect when the polarization direction of the normal incident light propagates along the *x* and *y* axes. To realize different *T_x_* and *T_y_* values, *L_x_* cannot be equal to *L_y_*. Considering the lossless properties of the scatterer material GaN and substrate material SiO_2_ in visible wavelengths, the sum of reflectivity *R* and transmittivity *T* are constant and equal 1. Based on the rotational symmetry of rectangles, we can realize the switch between *T_x_* and *T_y_* by means of rotating the scatterers by 90°, which converts *R_t_* to its reciprocal.

Using the professional software FDTD Solutions (Rsoft 2013, Montreal, QC, Canada), periodic boundary conditions (PBC) were set along both *x* and *y* directions, while perfectly matched layers were set along the *z* direction. The measurement of transmittivity *T* was performed on a constant air reference plane far enough away from the metasurface, and reflectance was calculated as *R* = 1 − *T*. In our simulation, we set both the scatterer height *h*_1_ and substrate *h*_2_ at 500 nm and the unit size *P* as 260 nm, i.e., smaller than the incident wavelength of 532 nm, to prevent the occurrence of grating diffraction. This setting is satisfied for high-efficiency backward reflections when electric or magnetic mode resonances occur. By tuning the two lengths, *L_x_* and *L_y_*, of the scatterers, we can find appropriate geometric parameters that satisfy the transmission conditions and realize the corresponding polarizer effect.

## 3. Results

### 3.1. Tunable Polarizer

We define *R_t_* as the ratio between *T_y_* and *T_x_*. To realize a tunable polarizer, we calculated *R_t_* with different geometrical parameters. From the sweep simulation results, as shown in Figure 2, with constant values of *L_x_* and varying values of *L_y_*, *T_x_* is larger than 90% in most regions (Figure 2b) while *T*_y_ decreases in the corresponding region (Figure 2a), which results in the a large scope for *R_t_*, as shown in Figure 2c. For evaluating the degree to which our tunable polarizer was optimized, we defined the resolution ratio, *R_r_*, which is the ratio between the ranges of *R_t_* and *L_y_*. As a result, we achieved a tunable polarizer with the parameter *R_t_* ranging from 0 to 1. For example, when *L*_x_ is constant at 138 nm and *L_y_* ranges from 180 to 220 nm, *R_t_* ranges from 0.00031 to 1, which produces a small *R_r_* of 4.3%/nm. When *L_y_* = 203 nm, *T_y_* is equal to its lowest value of 0.03%, and *R_t_* = 0.00031. When *L_x_* = 120 nm and *L_y_* varies from 180 to 240 nm, *R*_r_ is equal to its lowest value of 1.8%/nm, which is the most insensitive *R_r_* value in our design. In the case of *L_y_* = 235 nm, *T_y_* = 0.6% and *R_t_* = 0.00061. As *R_r_* decreases, the modulation of our tunable polarizer becomes easier to control and more practical. Since our unit cells are symmetric, the tunable polarizer effect can be reversed by rotation. For a geometrical configuration, such as when *L_x_* and *L_y_* are respectively equal to 120 and 235 nm, we achieve perfect reflection with all-dielectric metasurfaces, similar to reports in the literature [28]. However, we have derived selectable and perfect reflection due to the rotational symmetry of our structure, which is different from previous work, where the arrays had achieved polarization insensitivity with spherical scatterers. Moreover, our polarizer can act as a light splitter for *x*- and *y*-polarized, normally incident light because of the pure lossless dielectric materials used. As a result, the incident polarized light will propagate along the forward direction, and the other polarized beam will be reflected normally. Furthermore, for the tunable polarizer, the proportion of transmitted or reflected component can be controlled. Based on the polarization dependence of our design, *T_x_* and *T_y_* still present different characteristics. By observing the transmittivity spectrum for *x*- and *y*-polarized, normally incident light, where *L_x_* and *L_y_* are respectively equal to 120 and 235 nm, as shown in Figure 2d, the dips of *x* polarization do not coincide with the dips of *y* polarization. Additionally, at a wavelength of 532 nm, we observe that a polarizer with large *T_y_* and small *T_x_* values has a FWHM (full width at half maximum) value of around 3 nm. For our polarization-dependent structure, the dips in transmission spectrum for the *x*-polarized incidence do not coincide with those of *y* polarized incidence. The above phenomenon can be interpreted as a waveguide, where some modes are cut off and cause high reflectance [9,37,38].

To obtain in-depth knowledge on the physical mechanisms regarding polarizer effects, we have presented results from experiments using an electric and magnetic field intensity at an incident wavelength equal to 532 nm for *y*-polarized incidence with *L_x_* = 120 nm, *L_y_* = 235 nm and *L_x_* = 138 nm, *L_y_* = 203 nm, corresponding to the lowest *T_y_* and the minimum *R_t_,* respectively. As can be seen in Figure 3 and Figure 4, the distributions of electric and magnetic field magnitudes for the two aforementioned geometrical configurations are quite similar. By observing the distribution magnitudes of the electric field (Figure 3a,c), also according to the *z* component (Figure 3b,d), we find that the *z* component of the electric field occupies a principle proportion, and that the largest magnitude is located in the central area of the scatterers. Based on the magnitude distributions of the electric and magnetic field at the central scatterer area in the *x*–*y* plane, as seen in Figure 3a,e, a typical magnetic mode resonance of Mie resonances occurs, which produces high reflectance with dielectric metasurfaces and is in accordance with the work in [25]. Moreover, the field analysis of Figure 4 is the same as that of Figure 3.

Our design is completely composed of dielectric materials that have a narrow-band dip in the transmission spectrum for selectable incident polarization, which can act as a beam splitter. The use of our unit cell offers the possibility of a design with a color filter. Furthermore, our model has the potential to be used as an optical antenna for designing the metasurfaces with special light modulation. For instance, when circularly polarized light normally impinges on the metasurface, one polarization incidence is reflected while the other is transmitted with a non-zero refraction angle at high efficiency.

### 3.2. Application for Biosensing

For the above polarization-sensitive structure, *T_x_* and *T_y_* still present different characteristics with different environmental refractive indexes. When the transmission spectrum is within narrow-band and the resonant wavelength shifts are acceptably large, the transmit-array with polarization dependence acts as a high-performance biosensor. If the transmitted environment of metasurfaces with *L_x_* = 120 nm and *L_y_* = 235 nm was shifted from air to solution, we found that our polarizer performed well in biosensing applications for *y*-polarized incidence. For determining its sensing performance, factor of sensitivity *S* and *FOM* are defined as
(1)S=Δλ/Δn,
(2)FOM=S/FWHM,
where Δ*λ* and Δ*n* represent the spectral shift of resonant wavelength and variation of environmental refractive index, respectively [32,33,34]. When our metasurfaces were placed in a glucose solution environment [32] with a refractive index ranging from 1.30 to 1.39, the resonant wavelength has a redshift of 17.3 nm for the *y*-polarized, normally incident light, as shown in Fig. 5. According to Equations 1 and 2, we calculated a refractive index sensitivity *S* = 192 nm/RIU and *FOM* = 64/RIU for the proposed polarizer. We have found that our design has high performance within refractive index *n*, ranging from 1.3 to 1.39. For more accurate detection and practical application, the definition of *FOM** is widely used in biosensors, which is defined as
(3)$$FOM^{*}=max\left|\frac{dI\left(\lambda \right)/dn\left(\lambda \right)}{I\left(\lambda \right)}\right|,$$
where ∆*I*/*I* is the relative intensity variation due to the environmental refractive index variation ∆*n* at a specific wavelength *λ*, *I*(*λ*) refers to intensity, which is lower in comparison and, consequently, causes the maximum value of *FOM** [32,33,34]. From the transmission spectrum shown in Figure 5, the black dashed line is marked at the resonant wavelength 439.8 nm for a refractive index of 1.33, where the transmittivity reaches an extremely low value of 0.06%, and *FOM** reaches its maximum value, 92,333/RIU. The large *FOM** value is caused by the narrow-band feature being very narrow, and the dip in the transmission spectrum being extremely close to 0. For practical detection, the light intensity was measured with a different environmental refractive index surrounding each metasurface, and a fixed incident wavelength. Our metasurface, as a biosensor with extremely high *FOM**, shows great potential for applications in detection.

For more specific biosensing functions, we added a thin layer with thickness *l* to the GaN scatterers, which can act as the detected biomedical molecules. We set the environmental refractive index, *n* = 1.33, as the refractive index of water. Firstly, we set the refractive index of the thin layer as 1.5 and varied the thickness *l* from 1 to 20 nm at an incident wavelength of 503.5 nm. The corresponding transmittivity ranged from 0.96 to 0.1, as seen in Figure 6a, which clearly shows distinct transmittivity. The thickness, *l*, was set to 90 nm, with the refractive index of the thin layer ranging from 1.4 to 1.7. We also added the transmission spectrum without a thin layer for the environmental refractive index *n* = 1.33 to Figure 6b–d, which represents the refractive index of the thin layer equal to 1.33. We found that the *FOM** value for the thin-layered structure was 15167/RIU, which is related to high biosensing performance. In Figure 6c, all transmission spectrums within wavelengths 430–450 nm are similar, and the dips are very close. However, in Figure 6d, the resonant wavelengths have an obvious shift due to the variation of refractive index of the thin layer, and the FWHM is still narrow, which results in a high *FOM**. Based on the above performance parameters, our design can be operated as a biosensor, functioning in the visible regime and offers a novel application of pure dielectric metasurfaces—the sensing and detection of chemical reactions in nanoscale dimensions. For metasurfaces of lossless materials, the dip of the transmission spectrum represents the peak of the reflectance curve. Consequently, detection can be implemented with either the reflectance or transmission spectrums, which offers great detection convenience.

## 4. Conclusions

In summary, we have proposed a tunable polarizer for illumination under *x*- and *y*-polarized illumination and normal incidence, based on Mie resonances of dielectric materials in the visible regime. Our polarizer has a large scope of *R_t_*, ranging from 1% to 0.031%, with an insensitive resolution ratio of *R_r_* = 1.8%/nm. The narrow-band properties of the polarizer allow it to be utilized as a biosensor in biochemistry or life sciences, due to the superior performance using a refractive index of 0.09 in detection, acceptable sensitivity *S*, and an extremely large *FOM** value. Although our design is polarization-dependent, the transmit-array metasurface-based biosensor may be a potential market candidate in the future. The methodology of using narrow-band polarization sensitivity to design a high-performance biosensor can also be applied in reflect-array metasurfaces working in other wavelengths. Our developed transmit-array biosensor with polarization insensitivity shows great promise.

## Figures and Tables

**Figure 1 nanomaterials-09-00603-f001:**
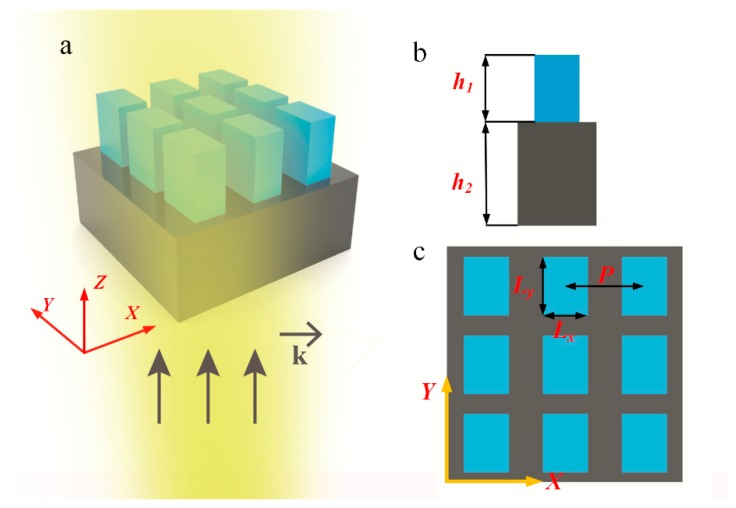
Geometry of the polarizer. (**a**) Sketch of arrays consisting of GaN rectangular posts and SiO_2_ substrate. (**b**) Schematic of the unit cell with a scatterer height of *h*_1_ and substrate height of *h*_2_. (**c**) Vertical view of arrays illustrating the period *P* and scatterer dimensions *L_x_* and *L_y_*.

**Figure 2 nanomaterials-09-00603-f002:**
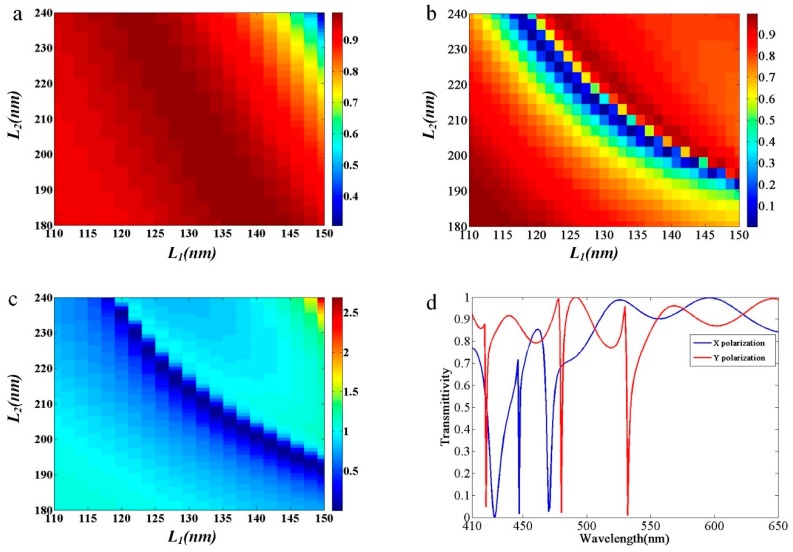
(**a**) Calculated transmittivity for the *y*-polarized incidence at a wavelength of 532 nm. (**b**) Calculated transmittivity for the *x*-polarized incidence. (**c**) Calculated *R_t_* distribution. (**d**) Transmittivity spectrum for *x*-polarized (blue line) and *y*-polarized (red line) incidence with lengths *L_x_* and *L_y_* respectively equal to 120 and 235 nm.

**Figure 3 nanomaterials-09-00603-f003:**
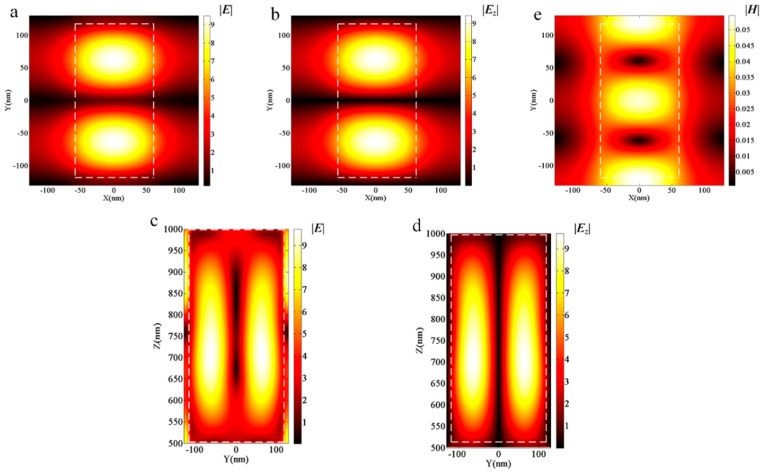
The field distribution of metasurfaces with *L_x_* = 120 nm and *L_y_* = 235 nm for the *y* polarized incidence. (**a**) The electric field distribution in the central *x*–*y* plane of GaN scatterers. (**b**) The *z* component distribution of the electric field in the central *x*–*y* plane of GaN scatterers. (**c**) The electric field distribution in the *y*–*z* plane, *x* = 0. (**d**) The *z* component distribution of the electric field in the *y*–*z* plane, *x* = 0. (**e**) The magnetic field distribution in the central *x*–*y* plane of GaN scatterers.

**Figure 4 nanomaterials-09-00603-f004:**
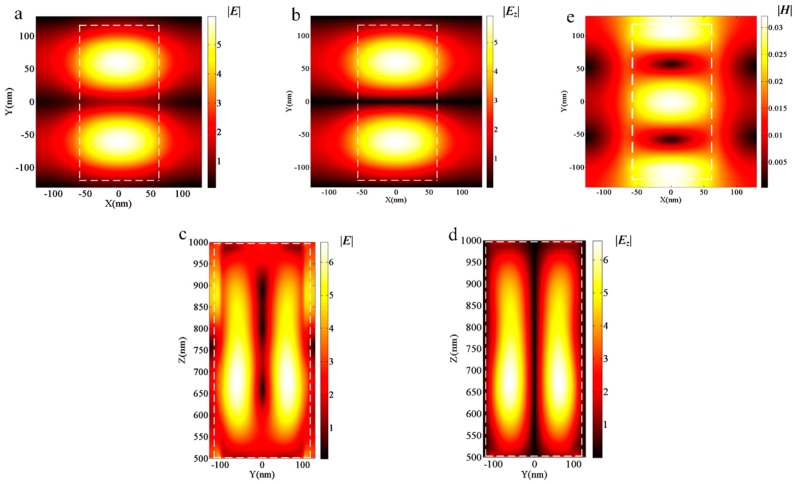
The field distribution of metasurfaces with *L_x_* = 138 nm and *L_y_* = 203 nm for the *y*-polarized incidence. (**a**) The electric field distribution in the central *x*–*y* plane of GaN scatterers. (**b**) The *z* component distribution of the electric field in the central *x*–*y* plane of GaN scatterers. (**c**) The electric field distribution in the *y*–*z* plane, *x* = 0. (**d**) The z-component distribution of the electric field in the *y*–*z* plane, *x* = 0. (**e**) The magnetic field distribution in the central *x*–*y* plane of GaN scatterers.

**Figure 5 nanomaterials-09-00603-f005:**
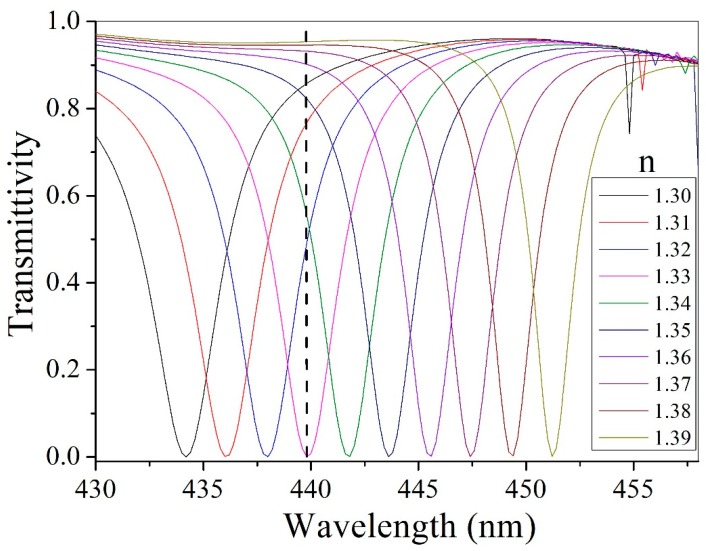
Simulated transmission spectrum of metasurfaces placed in environments with a refractive index ranging from 1.3 to 1.39. The black vertical dashed line is marked to calculate *FOM**.

**Figure 6 nanomaterials-09-00603-f006:**
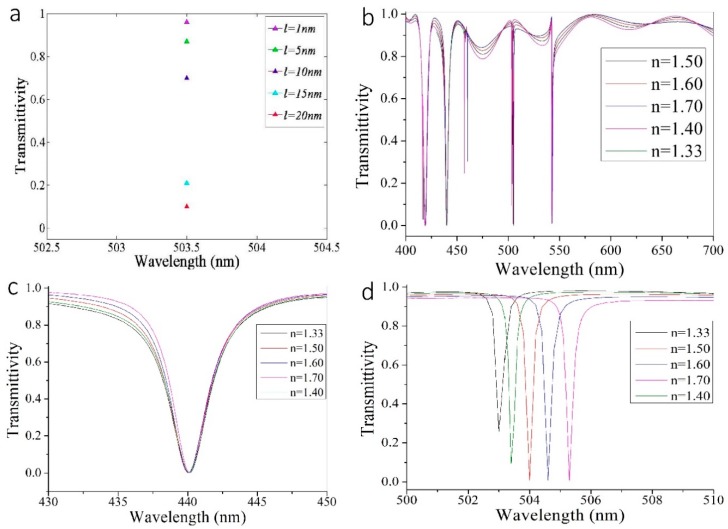
The transmission spectrum for thin layer with a constant environmental refractive index of 1.33. (**a**) The thin layer transmittivity for a constant refractive index of 1.5 and various thicknesses *l*. (**b**) The broadband transmission spectrum for thin layers with constant thickness *l* = 90 nm and various refractive indexes. (**c**) The specific transmission spectrum for thin layers with constant thickness *l* = 90 nm and various refractive indexes within a wavelength range of 430–450 nm. (**d**) The specific transmission spectrum for thin layers with constant thickness *l* = 90 nm and various refractive indexes within a wavelength range of 500–510 nm.

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
