# Peer review of "Transmit-Array, Metasurface-Based Tunable Polarizer and High-Performance Biosensor in the Visible Regime"

_nanomaterials, 2019, doi:10.3390/nano9040603_

Reviewer 1 Report

The paper is clear and pretty well written. The topic is very interesting.
On the other hand, some parts are too much descriptive and few rigorous.
In the following, my comments to address before acceptance:

_Section "1. Introduction"

The introduction is really clear and all the objectives well stated.
On the other hand some recent technology developments are missing such as:

_ nanoparticles [Electromagnetic nanoparticles for sensing and medical diagnostic applications, Materials 11 (4), 603, 2018]
_ near-zero-index materials [Near-zero refractive index photonics, Nature Photonics, 11, 149–158, 2017]
_ curvilinear surfaces [Curvilinear MetaSurfaces for Surface Wave Manipulation, Scientific Reports 9 (1), 3107, 2019]
_ graphene [Broadband, electrically tunable third-harmonic generation in graphene, Nature Nanotechnology 13, 583–588, 2018]
_ metamaterials [Metamaterial-based wideband electromagnetic wave absorber, Optics express 24 (6), 5763-5772, 2016]

It would be beneficial if authors can expand and explain such techniques and make a comparison among theirs and the above suggested works: pro/cons should be highlighted.

_ Section "2. Materials and Methods"

The related method is very well detailed and similar to [Metasurfaces for Advanced Sensing and Diagnostics, Sensors, 19(2), 355, 2019]. Therefore:

Explain in details what are the advantages/disadvantages and similarities/differences of your method compared to the above-mentioned.

_ Section "3. Results"

3.1) A comparison in terms of transmittivity (Fig.2) obtained from the analytical model above mentioned and your results, it would be beneficial for the reader and  a good proof-of-concept to confirm the reliability of your approach.

3.2) A deeper study of the single particle properties is needed.
3.2.1) Authors should write some lines in this paragraph to explain why they chose to use the proposed structure
3.2.2) Explore the capability of the shape and compare yours with the one contained in the above-mentioned work.
What are the advantages/disadvantages?

3.3) Authors should refer to the following additional phenomena:
_ electric/equivalent magnetic currents [New absorbing boundary conditions and analytical model for multilayered mushroom-type metamaterials: applications to wideband absorbers, IEEE Trans. Antenn. Propag., 60(12), 5727–5742, 2012]
_ surface waves [Near-zero-index wires, Optics express 25 (20), 23699-23708, 2017]
_ displacement currents [Hybrid bilayer plasmonic metasurface efficiently manipulates visible light, Science Advances, 2(1), 2016]

How such phenomena affect the structure functionality and in particular its properties?

3.4) The paper lack in further application examples. Besides the sensing, other applications can be taken into account, such as: refractive index measurements, diagnostics, nano-therapy, nanocommunications.
I would suggest to consider such applications and explaining how you can use your device for them.
Please highlight what's new in yours.
A specific paragraph would be very useful for the reader.

_ Section "4. Conclusions"

1) No limitations of the proposed method have been highlighted.

2) No future improvements/works have been discussed.

Author Response

Response to Reviewer 1 Comments

Point 1: The introduction is really clear and all the objectives well stated. 
On the other hand, some recent technology developments are missing such as:

_ nanoparticles [Electromagnetic nanoparticles for sensing and medical diagnostic applications, Materials 11 (4), 603, 2018]

 _ near-zero-index materials [Near-zero refractive index photonics, Nature Photonics, 11, 149–158, 2017]

 _ curvilinear surfaces [Curvilinear MetaSurfaces for Surface Wave Manipulation, Scientific Reports 9 (1), 3107, 2019]

 _ graphene [Broadband, electrically tunable third-harmonic generation in graphene, Nature Nanotechnology 13, 583–588, 2018]

 _ metamaterials [Metamaterial-based wideband electromagnetic wave absorber, Optics express 24 (6), 5763-5772, 2016]

It would be beneficial if authors can expand and explain such techniques and make a comparison among theirs and the above suggested works: pro/cons should be highlighted.

Response 1:  Metasurfaces, offer us more and more interesting techniques such as above literatures [1-8]. Comparing with above works, our structure is composed of all-dielectric materials and without metals. Other designs of biosensors demand metals, which means the existence of absorption (A). With lossless materials, whatever the geometric parameters we use, the relation between transmissivity(T) and reflectance(R) is simple: R=1-T. Consequently, the detection function can be processed with transmission spectrum and reflectance spectrum. When absorption (A) is not zero and a function of incident wavelength, the detection only can be detected by one choice of transmission or reflectance spectrum. The performance of biosensors can be represented by different factors, and the most practical factor is FOM*. Based on our design, we realized a very high FOM* 92333 in the refractive index from 1.30 to 1.39. And our design is polarization sensitive, which can be improved in the future. Thanks for providing us so many interesting papers [1-8] about the new technology about metasurfaces and biosensors. And we will expand our references with them.

Point 2: The related method is very well detailed and similar to [Metasurfaces for Advanced Sensing and Diagnostics, Sensors, 19(2), 355, 2019]. Therefore:

Explain in details what are the advantages/disadvantages and similarities/differences of your method compared to the above-mentioned.

Response 2: The work in the literature [8] presents different structures to act as a biosensor, which include reflective and transmitted models. Our work and their work both used the transmission spectrum to realize the biosensing application. Whatever the reflective and transmitted structures their work has, their structures contain metal patches. However, our structure is completely composed of lossless dielectric materials. The existence of metal means the energy losses, and the distribution of incident energy will be divided into absorption and scattering (transmission and reflection). As a result, the detection of refraction index can only be detected by the one choice of reflectance spectrum and transmission spectrum. But our counterpart is composed of all dielectric materials. Consequently, the detection of refraction index can be detected by the single choice of reflectance spectrum and transmission spectrum.

Point 3: 3.1) A comparison in terms of transmissivity (Fig.2) obtained from the analytical model above mentioned and your results, it would be beneficial for the reader and a good proof-of-concept to confirm the reliability of your approach.

Response 3: We defined Rt as the ratio between Ty and Tx. To realize a tunable polarizer, we calculated Ty, Tx and Rt with different geometrical parameters as shown in Fig.2 of the uploaded manuscript. As a result, we realized a tunable polarizer with Rt from 0 to 1. For example, when L1 is constant at 138nm and L2 is ranging from 180 nm to 220 nm, Rt is ranging from 0.031% to 1.

Point 4:

3.2.1) Authors should write some lines in this paragraph to explain why they chose to use the proposed structure 

3.2.2) Explore the capability of the shape and compare yours with the one contained in the above-mentioned work. 

 What are the advantages/disadvantages?

Response 4:

3.2.1) When we set L1=L2=L, the transmission spectrum about L in the free space is shown as the Fig .1 in this response letter. Only When L=176nm, a dip of transmission spectrum occurs and the transmissivity T is equal to its lowest value 0.008. Then we change the surrounding refractive index to detect the sensing performance with this geometric configuration L1=L2=176nm. However, the performance is not satisfying. So we cannot design a high-performance biosensor with polarization insensitivity (symmetry structure). So we used the asymmetry geometric configuration of (L1, L2) equal to (120nm,235nm) to realize a high-performance biosensor. And the sensing performance of other asymmetry geometric configuration also has been detected, which cannot catch up with the high performance of the geometric configuration of (120nm,235nm).

Fig .1 Transmittivity spectrum about L

3.2.2) We use different shape and materials compared with the work in the literature [8]. And the mechanism of their work is not same as our work is not same. We used the Mie resonances of dielectric materials, and they used the plasmonic resonances of metal. As for performance, the sensitivity factor S of their biosensor is more high than ours. However, they don’t calculate the more important factor FOM*. And we realized a high FOM*, which is higher than the work in the literatures [16-19]. In some references about biosensors in the uploaded manuscript, FOM* is not calculated. The higher the FOM* is, the more practical the biosensor is. And the reason is that the dip of reflectance spectrum is very close to 0, which is equal to 0.06%.

Point 5:

3.3) Authors should refer to the following additional phenomena: 
_ electric/equivalent magnetic currents [New absorbing boundary conditions and analytical model for multilayered mushroom-type metamaterials: applications to wideband absorbers, IEEE Trans. Antenn. Propag., 60(12), 5727–5742, 2012]
_ surface waves [Near-zero-index wires, Optics express 25 (20), 23699-23708, 2017]
_ displacement currents [Hybrid bilayer plasmonic metasurface efficiently manipulates visible light, Science Advances, 2(1), 2016]

How such phenomena affect the structure functionality and in particular its properties?

Response 5:

In our work, we used the dielectric scatterers to realize the biosensing application. In previous researches [9-15], it has been widely studied how the concerned physical quantities owning to the variation of geometrical parameters affect the transmission coefficient of the transmit-array metasurfacs, such as the diameter of nanodisks and the distance between scatterers. And The surface plasmonic waves are the special property of metal, which are excited in the surface of metallic particles. The Mie resonances are the special property of lossless dielectric materials, which are excited in the center of the dielectric particles. Owing to the all-dielectric structure, we don’t use too much text to study how the above phenomena affect the structure functionality. The study of special surface waves usually accompanies with metal structure. However, it offers us more information to improve our introduction as a surface wave manipulation application of metasurfaces.

Point 6:

3.4) The paper lack in further application examples. Besides the sensing, other applications can be taken into account, such as: refractive index measurements, diagnostics, nano-therapy, nanocommunications.

Response 6:

Our design composed of all-dielectric materials has a narrow-band dip in the transmission spectrum for specific incident polarization, which can act as a beam splitter. The use of our unit cell offers a possible way to design a colour filter. Furthermore, our design is potential to be used as the optical antenna to design metasurfaces with special light modulation. For example, when a circularly polarized incident light normally impinges on one metasurface, one polarization of incidence is reflected and the other polarization is high-efficiency transmitted with a non-zero refraction angle.

Point 7: 4.1) No limitations of the proposed method have been highlighted.

Response 7:

Our design is polarization-dependence.

Point 8: 4.2) No future improvements/works have been discussed.

Response 8:

In the future, we can design an all-dielectric biosensor with polarization insensitivity.

References

1.          La Spada, L.; Vegni, L. Metamaterial-based wideband electromagnetic wave absorber. Optics Express 2016, 24, 5763–5772.

2.          La Spada, L.; Vegni, L. Electromagnetic Nanoparticles for Sensing and Medical Diagnostic Applications. Materials 2018, 11, 603.

3.          La Spada, L.; Vegni, L. Near-zero-index wires. Opt. Express 2017, 25, 23699–23708.

4.          La Spada, L.; Spooner, C.; Haq, S.; Hao, Y. Curvilinear MetaSurfaces for Surface Wave Manipulation Scientific Reports 2019, 9, 3107.

5.          Qin, F.; Ding, L.; Zhang L.; et al. Hybrid bilayer plasmonic metasurface efficiently manipulates visible light. Science Advances 2016, 2.

6.          Liberal, I.; Engheta, N. Near-zero refractive index photonics. Nature Photonics 2017, 11, 149. 

7.          Soavi, G.; Wang, G.; Rostami, H.; et. al. Broadband, electrically tunable third-harmonic generation in graphene. Nature Nanotechnology 2018, 13, 583.

8.          La Spada, L. Metasurfaces for Advanced Sensing and Diagnostics. Sensors 2019, 19, 355.

9.          Cheng, J. R.; Ansari-Oghol-Beig, D.; Mosallaei, H. Wave manipulation with designer dielectric metasurfaces. Opt. Lett. 2014, 39, 6285-6288.

10.       Evlyukhin, A. B.; Reinhardt, C.; Seidel, A.; et al. Optical response features of Si-nanoparticle arrays. Physical Review B 2010, 82, 045404.

11.       Decker, M.; Staude, I.; Falkner, M.; et al. High-Efficiency Dielectric Huygens' Surfaces. Advanced Optical Materials 2015, 3, 813-820.

12.       Ahmadi, A.; Mosallaei, H. Physical configuration and performance modeling of all-dielectric metamaterials. Phys. Rev. B. 2008, 77, 045104.

13.       Decker, M.; Staude I. Resonant dielectric nanostructures: a low-loss platform for functional nanophotonics. J. Opt. 2016, 18, 103001.

14.       Evlyukhin, A. B.; Reinhardt, C; Chichkov, B. N. Multipole light scattering by nonspherical nanoparticles in the discrete dipole approximation. Phys. Rev. B 2011, 84, 235429.

15.       Staude, I.; Miroshnichenko, A. E.; Decker, M.; Fofang, N. T.; Liu, S.; Gonzales, E.; Dominguez, J.; et al. Tailoring directional scattering through magnetic and electric resonances in subwavelength silicon nanodisks. ACS Nano 2016, 7, 7824–7832. 

16.       Liu, N.; Mesch, M.; Weiss, T.; Hentschel, M.; Giessen, H. Infrared perfect absorber and its application as plasmonic sensor. Nano Lett. 2010, 10, 2342–2348. 

17.       Becker, J.; Trugler, A.; Jakab, A.; Hohenester, U.; Sonnichsen, C. The optimal aspect ratio of gold nanorods for plasmonic bio-sensing. Plasmonics. 2010, 5, 161–167.

18.       Lu, X. Y.; Wan, R. G..; Zhang, T. Y. Metal-dielectric-metal based narrow band absorber for sensing applications. Optics Express 2015, 23, 29842-29847.

19.       Lu, X. Y.; Zhang, T. Y.; Wan, R. G. Nanoslit-microcavity-based narrow band absorber for sensing applications. Optics Express 2015, 23, 20715-20720.

Reviewer 2 Report

Transmit-array metasurfaces-based tunable polarizer and high-performance biosensor in visible regime

MDPI nanomaterials March 2019

In this paper, ONLY simulations of a dielectric metasurface composed of periodically ordered nanobricks made of GaN placed on top of a glass substrate is designed/analyzed. Furthermore, one specific design is analyzed spectrally in order to see its sensitivity to refractive index changes between 1.3 and 1.39 (glucose).
Unfortunately, NO fabrication and experimental analysis is presented. Therefore the overall substance of this paper is quite limited, however very interesting and promising. With the present electromagnetic (EM) content, I suggest paper is revised and experimental results included.
I don’t find the title correct as no bio… has been included, only a theoretical variation of the refractive index (refractometer).
If we the focus on the EM calculation presented in figs. 2 and 3. Then, it is not clear why in fig. 2c the Rt is larger than 1 and from fig. 3 it is not clear what kind of resonances are present.
How sensitive is the sensor to variations in the angle of incidence of the incoming light?
My impression is also that the reference list needs widening.

Author Response

Response to Reviewer 2 Comments

Point 1:  In this paper, only simulations of a dielectric metasurface composed of periodically ordered nanobricks made of GaN placed on top of a glass substrate is designed/analyzed. Furthermore, one specific design is analyzed spectrally in order to see its sensitivity to refractive index changes between 1.3 and 1.39 (glucose).
Unfortunately, NO fabrication and experimental analysis is presented. Therefore the overall substance of this paper is quite limited, however very interesting and promising. With the present electromagnetic (EM) content, I suggest paper is revised and experimental results included.

Response 1: Thanks for your comments and suggestions. The experimental results are surely important. However, because of the limited condition of facility in our laboratory, currently, we cannot supplement the experimental results in this manuscript.

Point 2:  I don’t find the title correct as no bio… has been included, only a theoretical variation of the refractive index (refractometer).

Response 2:  We are sorry for that we don’t have the facility to perform an experiment. So we only can use the different refractive index in simulation to replace the chemical solutions with different concentration.

Point 3: If we the focus on the EM calculation presented in figs. 2 and 3. Then, it is not clear why in fig. 2c the Rt is larger than 1 and from fig. 3

Response 3:  We defined Rt as the ratio between Ty and Tx. To realize a tunable polarizer, we calculated Ty, Tx and Rt with different geometrical parameters as shown in Fig.2 of the uploaded manuscript. As a result, we realized a tunable polarizer with Rt from 0 to 1. For example, when L1 is constant at 138nm and L2 is ranging from 180 nm to 220 nm, Rt is ranging from 0.031% to 1. And in fig.2c some values of Rt are larger than 1, which means the corresponding geometrical parameters are not useful for a tunable polarizer actually.

Point 4: It is not clear what kind of resonances are present.

Response 4:  We used the Mie resonances of dielectric materials to design a polarizer such as in fig. 3 of the uploaded manuscript. The study of Mie resonances in different literatures [1-7] adopted different physical quantity to analyze the field distribution. We used the mode analysis as the literature [7]. When we used the vector analysis to study the filed distribution as shown in Fig. 1 below, we can find a magnetic dipole resonance, and the magnetic field lines are enclosed.

Fig. 1 the vector diagram of magnetic field in the center of the GaN scatterers.

Point 5: How sensitive is the sensor to variations in the angle of incidence of the incoming light?
Response 5:  In the literatures [8-9], the relation between transmissivity of transmit-array and incident angle has been studied in detail, which is that transmissivity will reach its largest value with the normal incident angle. And the study of performance of biosensors almost pay attention to the normal incidence [10-16]. Considering this, we think it is not necessary to re-study how the sensor is sensitive to the incident angle in this manuscript.

Point 6: My impression is also that the reference list needs widening.

Response 6:  We have expanded our references with the literatures [1-3,15-24], which is about the new technology about metasurfaces and biosensors.

References

1.          Cheng, J. R.; Ansari-Oghol-Beig, D.; Mosallaei, H. Wave manipulation with designer dielectric metasurfaces. Opt. Lett. 2014, 39, 6285-6288.

2.          Evlyukhin, A. B.; Reinhardt, C.; Seidel, A.; Luk'yanchuk, B.; Chichkov, B. Optical response features of Si-nanoparticle arrays. Physical Review B 2010, 82, 045404.

3.          Decker, M.; Staude, I.; Falkner, M.; et al. High-Efficiency Dielectric Huygens' Surfaces. Advanced Optical Materials 2015, 3, 813-820.

4.          Ahmadi, A.; Mosallaei, H. Physical configuration and performance modeling of all-dielectric metamaterials. Phys. Rev. B. 2008, 77, 045104.

5.          Decker, M.; Staude I. Resonant dielectric nanostructures: a low-loss platform for functional nanophotonics. J. Opt. 2016, 18, 103001.

6.          Evlyukhin, A. B.; Reinhardt, C; Chichkov, B. N. Multipole light scattering by nonspherical nanoparticles in the discrete dipole approximation. Phys. Rev. B 2011, 84, 235429.

7.          Staude, I.; Miroshnichenko, A. E.; Decker, M.; Fofang, N. T.; Liu, S.; Gonzales, E.; Dominguez, J.; et al. Tailoring directional scattering through magnetic and electric resonances in subwavelength silicon nanodisks. ACS Nano 2016, 7, 7824–7832. 

8.          Ebbesen, T. W.; Lezec, H. J.; Ghaemi, H. F.; Thio, T.; Wolff, P. A. Extraordinary optical transmission through sub-wavelength hole arrays. Nature 1998, 391, 667-669.

9.          Lezec, H. J.; Degiron, A.; Devaux, E.; et al. Beaming light from a subwavelength aperture. Science 2002, 297, 820-822.

10.       Lu, X. Y.; Zhang, T. Y.; Wan, R. G.; Xu, Y. T.; Zhao, C. H. Numerical investigation of narrowband infrared absorber and sensor based on dielectric-metal metasurface. Opt. Express 2016, 26, 10179-10187.

11.       Liu, N.; Mesch, M.; Weiss, T.; Hentschel, M.; Giessen, H. Infrared perfect absorber and its application as plasmonic sensor. Nano Lett. 2010, 10, 2342–2348. 

12.       Becker, J.; Trugler, A.; Jakab, A.; Hohenester, U.; Sonnichsen, C. The optimal aspect ratio of gold nanorods for plasmonic bio-sensing. Plasmonics. 2010, 5, 161–167.

13.       Ameling, R.; Langguth, L.; Hentschel, M.; Mesch, M.; Braun, P. V.; Giessen, H. Cavity-enhanced localized plasmon resonance sensing. Appl. Phys. Lett2010, 97, 253116.

14.       Cetin, A. E.; Altug, H. Fano resonant ring/disk plasmonic nanocavities on conducting substrates for advanced biosensing. ACS Nano 2012, 6, 9989–9995.

15.       Lu, X. Y.; Wan, R. G.; Zhang, T. Y. Metal-dielectric-metal based narrow band absorber for sensing applications. Optics Express 2015, 23, 29842-29847.

16.       Lu, X. Y.; Zhang, T. Y.; Wan, R. G. Nanoslit-microcavity-based narrow band absorber for sensing applications. Optics Express 2015, 23, 20715-20720.

17.       La Spada, L.; Vegni, L. Metamaterial-based wideband electromagnetic wave absorber. Optics Express 2016, 24, 5763–5772.

18.       La Spada, L.; Vegni, L. Electromagnetic Nanoparticles for Sensing and Medical Diagnostic Applications. Materials 2018, 11, 603.

19.       La Spada, L.; Vegni, L. Near-zero-index wires. Opt. Express 2017, 25, 23699–23708.

20.       La Spada, L.; Spooner, C.; Haq, S.; Hao, Y. Curvilinear MetaSurfaces for Surface Wave Manipulation Scientific Reports 2019, 9, 3107.

21.       Qin, F.; Ding, L.; Zhang L.; et al. Hybrid bilayer plasmonic metasurface efficiently manipulates visible light. Science Advances 2016, 2.

22.       Liberal, I.; Engheta, N. Near-zero refractive index photonics. Nature Photonics 2017, 11, 149. 

23.       Soavi, G.; Wang, G.; Rostami, H.; et. al. Broadband, electrically tunable third-harmonic generation in graphene. Nature Nanotechnology 2018, 13, 583.

24.       La Spada, L. Metasurfaces for Advanced Sensing and Diagnostics. Sensors 2019, 19, 355.

Round  2

Reviewer 1 Report

Authors answered properly to reviewer's questions.
New interesting applications and future works can be envisioned.

Author Response

Thanks for your reviewing. The comments are helpful for improving both technical and editing quality of this manuscript.

Regarding your comments, we checked English throughout the manuscript. Some incorrect expressions and words are corrected accordingly. Some paragraphs are polished.

Reviewer 2 Report

Dear all

I´ll maintain my dessision from the frist review round, that the papir is lacking experimental results, and the title therefore also is misleading.

The performed EM calculations is OK, but still needs some experimental evidence.

Author Response

Thanks for your comments.

Regarding English language and style are fine/minor spell, we surely found a lot of expression mistakes and word spell problems. We corrected them one-by-one already. After that, we thoroughly checked and polished English throughout the manuscript.

Regardsing the experiment issue, it is surely too difficult to arrange the experiment due to limitation of our current facility condition. But the manuscript is well written. It presents our novel idea. And the results are significant and meaningful. We think it is worth for publication in its current presentation form.